# Price Stability Properties and Volatility Analysis of Precious Metals: An ICSS Algorithm Approach

Sameen Fatima [1], Christopher Gan [1] and Baiding Hu [2,*]

1    Department of Financial and Business Systems, Lincoln University, Christchurch 7647, New Zealand
2    Department of Global Value Chains and Trade, Lincoln University, Christchurch 7647, New Zealand
*    Correspondence: baiding.hu@lincoln.ac.nz

**Abstract:** This paper investigates the price stability properties of precious metals during the 1997 Asian Financial Crisis, 2007–2008 Global Financial Crisis, and 2010 Eurozone Crisis. To analyse the interaction between precious metal prices and the US stock market stock performances, we use the ICSS algorithm along with the GARCH model to evaluate how the number of rapid changes in volatility of precious metals has been reduced. The results suggest gold is the most stable of the precious metals. However, silver, platinum, and palladium showed positive price correlation when the US Dow Jones market was unstable. These results imply that: (1) the correlation among stocks market returns has little to no significant impact on the price movement of precious metals, but the US Dow Jones has some influence on precious metal markets except gold, which means investors can reap this benefit from diversification; (2) investors can systematically increase their portfolio returns by going short with the gold investments with low price co-movement and long on silver, platinum, and palladium with high co-movement with stock prices.

**Keywords:** stock market; precious metals; ICSS algorithm; GARCH; spillovers

## 1. Introduction

The relationships and spillovers among precious metals, specifically gold and other precious metals, have been widely studied. For example, Escribano and Granger (1998) compared the performance returns of gold and silver and found a strong simultaneous relationship in returns between them. This finding contradicts Ciner's (2001) results. Ciner studied gold and silver returns for future contracts traded in Tokyo from 1992 to 1998 and found no evidence of long-run linkages between the gold and silver price movements. Lucey and Tully (2006) concluded that the gold-silver parity weakened during the 1990s. In general, Kearney and Lombra (2008) found a negative relationship between gold and platinum. Studying the wider perspective of the relationships among precious metals, Batten et al. (2010) found that the volatility of individual precious metals was influenced by other precious metals from 1996 to 2006. Following Batten et al.'s results, Batten et al. (2014) identified the existence of constant spillovers between silver and gold. The findings differed for the other two precious metals (platinum and palladium). Finally, Antonakakis and Kizys (2015) suggested that the 2008 Global Financial Crisis weakened the spillovers of gold while strengthening platinum spillovers. Precious metals' prices (in particular, gold and silver) have significantly increased from 1991 to 2011. The significant increase in price is because of a number of factors, such as economic and financial crises, inflation expectations, and increased demand in emerging markets (Lee and Lin 2010). This leads investors to consider investing in the precious metals market; it may play a vital role in the diversification of portfolios Adrangi et al. (2003); Lucey and Tully (2006). Speculators and arbitrageurs pay close attention to spillover effects between precious metals and stock markets across other asset classes (i.e., the foreign exchange market and stock markets) and other countries because this is where they can make money.

We recognize that stock markets can be impacted by shocks in financial market uncertainties, while precious metal markets are steadier and secure resources that will not be highly impacted by outside shocks. The current paper's major contribution is the analysis of the price stability properties of the precious metal markets among the precious metals by taking into account structural breaks in the underlying volatility generating processes for the precious metal markets. The paper adds to the literature on the relationships and co-movements among the precious metals, since there is a surprising lack of research dealing with co-movements among precious metals during the financial market instability. This paper also highlights the extent of precious metals' returns co-movements along with the most stable metals in terms of price fluctuation, which provides additional empirical evidence for the discussion of whether gold is a safe haven against stock markets.

The finding in the present research would be a useful observation for investors to plan their speculation by investing precious metals in their portfolios. The plan of the paper is as follows. The Section 2 provides a literature review on studies on precious metal markets and their interactions with stock markets, especially during financial upheavals. Section 3 describes the data and the modelling techniques adopted in the present research, namely, the ICSS and its combination with GARCH. Section 4 discusses the empirical results, with conclusions contained in Section 5.

## 2. Literature Review

Numerous studies have shown that gold is a safe bet during periods of financial distress. For example, Baur and McDermott (2010) investigated the role of gold as a safe haven[1] against global financial turmoil. The authors studied gold as a haven against stock markets among major developed and emerging countries from 1979 to 2009. Baur and McDermott (2010) studied whether gold correlates with the bond and stock markets and whether gold can be a safe haven and/or a hedge for investor portfolios. Studies discussing this topic are relatively scarce. One group of authors examined the influence and the nature of the gold market (e.g., Baur and Lucey 2010; Capie et al. 2005; Faff and Hillier 2004; Faugere and Erlach 2005; Hillier et al. 2006; Sherman 1982) and others examined the safe haven properties of gold (see Baur and Lucey 2010; Upper 2000; Kaul and Sapp 2006). There appears to be only one study that explicitly examines the role of gold as a hedge against the US dollar (see Capie et al. 2005). Capie et al. (2005) studied the properties of four precious metals (gold, silver, platinum, and palladium) in a time varying manner. Chua et al. (1990) and Jaffe (1989) investigated the properties of gold in terms of hedging and portfolio diversification. The authors found that gold and gold stocks can be used to reduce portfolio risk. Their results showed that gold bullion could be used as an alternative investment strategy.

Several other scholars have studied other precious metals' hedging abilities and their correlation with the stock indices to identify their role in hedging against natural inflation. For example, Hillier et al. (2006) examined the role of precious metals in an investment portfolio. The authors used daily data for gold, silver, and platinum from 1976 to 2004. They found low correlations between precious metals and stock market indices and concluded that precious metals can provide opportunities for portfolio diversification. They also found that precious metals have hedging abilities, particularly during periods of "abnormal" stock market volatility. To expand on the idea of using precious metals as hedges against inflation, Ghosh et al. (1999) and Heidorn and Demidova-Menzel (2008) examined the determinants of the gold market and its characteristics as a channel for hedging against inflation.

However, not all studies agree that gold is a safe haven against stock markets. Baur and Lucey (2010) found that gold acted as a hedge against stock market indices on average and is a safe haven in extremely distressed financial market conditions for a very short time such as 15 days of trading. Baur and McDermott (2010) examined all major world stock market indices. They concluded that gold is both a safe haven and a hedge in Europe and the US, but not in Canada, Australia, Japan, and large emerging markets such as the BRIC countries (Brazil, Russia, India, and China). Coudert and Raymond (2010) examined

the US stock market during the recession of 2008 to 2009. They found that though in the short-term, gold provides a weak safe haven, it does not work in the long-term.

Batten et al. (2010) claimed that economic drivers and linkages among the precious metals (gold, silver, platinum, and palladium) are clearly distinct from each other. Gorton and Rouwenhorst (2019) based their study on previous research that argued that the commodity market can be best viewed as a single asset class that has advantageous risk-return capabilities and provides portfolio diversification. However, Batten et al. (2010) concluded that all the precious metals, except for silver, show conditional volatility from all variables (financial and monetary). This suggests that, even in this homogeneity of market, the same factors (monetary and financial) may not be significant for all precious metals. This supports the argument of Erb and Harvey (2019) and Scherer and He (2008) that precious metals (gold, silver, platinum, and palladium) are clearly distinct from each other, a claim that contradicts Gorton and Rouwenhorst's (2019) findings.

### 2.1. Precious Metals during the 1997 Asian Financial Crisis

Volatility spillover in international markets is an important research topic, not only for policy makers and researchers, but also for investors. This issue has a number of practical implications in terms of risk hedging, asset allocation, portfolio risk management, and market efficiency. However, the effects of the transmission of risk (both risk and returns) are more apparent during periods of financial turmoil. In fact, linkages among cross-markets increased sharply during the three financial market crashes, indicating the existence of contagion among the different financial markets and different asset classes.

Morales (2008) determined the volatility spillover of precious metals from 1997 to 2007. The author used the GARCH and EGARCH models to investigate the behaviour of precious metals during the 1997 Asian Financial Crisis and found evidence that depreciation of the precious metals under stable and unstable financial market conditions tended to increase market volatility and the mean stock returns for precious metals. Morales (2008) suggested that there is clear evidence of a volatility spillover running in a bidirectional way.

### 2.2. Precious Metals during the 2008 Global Financial Crisis

Following the 2008 Global Financial Crisis, the concept of "flight to quality" drew the attention of policymakers and investors alike. This gave investors and portfolio managers reasons to add precious metals to their portfolios as an alternative avenue for diversification, especially after the increased presence of financial investors and financialisaton of commodity markets. In fact, over the past decade, commodity futures have become one of the most attractive asset classes and an investment vehicle for the portfolio managers, just like bonds and stocks. Previous literature (Baur and Lucey 2010; Baur and McDermott 2010; Gürgün and Ünalmış 2014; Mensi et al. 2014) empirically studied safe haven properties of precious metals. They concluded that gold can be used as a hedge in financial markets.

Mensi et al. (2017) examined the time-varying risk spillover between precious metals (gold, silver, platinum, and palladium) and four major stock markets (the US, Europe, Asia, and Japan). They used the Diebold and Yilmaz (2012) spillover index. In their preliminary analysis, the authors found some periods of significant price fluctuations for all the stock markets. Among the precious metals, silver was the most volatile market during their study period (2000 to 2016). All series exhibit volatility fluctuations in their evolution during from 2008 to 2009. Most interestingly, silver showed significant price increases between 2010 and 2012, followed by a plunge after the 2010 Eurozone Crisis. The authors found significant volatility clustering for all the series of returns during the 2008 Global Financial Crisis.

Hillier et al. (2006) investigated the role of precious metals in the financial markets using the daily price data for gold, silver, and platinum from 1976 to 2004. According to the authors, following the collapse of the stock markets during the 2008 Global Financial Crisis, in addition to increased global financial market integration of bond and stock markets, investors increased their participation in precious metal markets, which resulted in price increases. They also found that precious metals other than gold had a

low correlation with stock market indices, particularly during periods of global financial market distress. Therefore, investment portfolios with precious metal exposure performed significantly better.

### 2.3. Precious Metals during the 2010 Eurozone Crisis

The financial markets did not expect that Greece's significant debt problems could trigger a European debt crisis. The weakening of government finances after the 2008 Global Financial Crisis resulted in a sudden loss of trust and confidence in both stock markets and sovereign debt, as well as driving alternative investment prices, such as precious metals, to record highs. The strong performance of precious metals, most commonly gold, during the economic downturn and especially during the 2010 Eurozone Crisis, led researchers to examine the properties and characteristics of these asset classes and their role in the global financial systems.

A number of studies such as Jaffe (1989) and Hillier et al. (2006) investigated the role of metals in diversifying investment portfolios. Uddin et al. (2019) examined both the spillover effect and price volatility of precious metals (gold, silver, platinum, and palladium) from 1999 to 2019. They wanted to investigate economic and fundamental events during the 2010 Eurozone Crisis. The authors found evidence of time varying and homogenous asymmetric spillover between the return volatilities of the precious metals. This suggests similarities in the cyclical relationship of precious metals with global and local fundamentals. They found evidence of the effects of the 2008 Global Financial Crisis and the 2010 Eurozone Crisis on all precious metal commodities. Uddin et al. (2019) also found that the trends in asymmetric spillover reached their highest from 2010 to the end of 2014's last quarter. This increase was associated with growing uncertainty caused by the 2010 Crisis.

The existence of linkages among advanced financial markets has been previously well documented and studied. Researchers investigating dynamic market linkages, which provide evidence of causal relationships, have also found significant volatility spillovers and price volatility across advanced markets (Bae et al. 2000; Hamao et al. 1990; Koutmos and Booth 1995; Theodossiou and Lee 1993).

In the last few decades, there has been growth and development in global financial markets characterised by increased capital movement and international trade across borders. These features have led to the integration and co-movement of individual financial markets. As a result, stock markets in one country can be affected by apparent fluctuations in the financial markets of another country, affecting the former's performance and trends. All stock market returns are not only influenced by their past performance, but also by global news from other international stock markets (Lin et al. 1994).

## 3. Data and Methodology

The data consist of weekly closing values for each precious metal. We selected the US$/Troy ounce for gold, the London Free Market price in US$/Troy ounce for platinum, the London Free Market US$/Troy ounce for palladium, and the Zurich silver price in US$/kilogram for silver. This gives us 3240 observations for all four precious metals market indices from 1 January 1997 to 31 December 2018; i.e., gold (the XAU Index), silver (the XAG Index), platinum (the XPT Index), and palladium (the XPD Index).

### 3.1. The Iterative Cumulative Sum of Squares (ICSS) Approach

The iterative cumulative sum of squares algorithm (ICSS) was designed by Inclán and Tiao (1994). It allows the detection of multiple breaks in the variance of time series data. However, the literature shows that it can overstate the number of actual breaks in the variance (Fernández 2020). A study by Dubois and Bacmann (2002) showed that the ICSS algorithm is questionable when there is conditional heteroscedasticity. The authors claimed that this problem can be resolved by filtering the time series data with the GARCH (1,1) model, then applying the ICSS algorithm to the standardised residuals. For comparison

purposes, this paper incorporates the tests for volatility shifts before and after filtering the time series data for serial correlation and conditional heteroscedasticity.

The ICSS algorithm assumes that the data always displays stationary variance over the start of the period until a sudden change takes place, resulting in a sequence of events, then the variance reverts to the stationary condition until the next sudden change event. This process repeats through time, regenerating observations in time series with an unknown number and amount of changes in the variances Sayani (2011).

To estimate the number of sudden changes in events in the time series data, we use the cumulative sum of square residuals. This is given by Equation (1):

$$C_k = \sum_{t=1}^{k} e^2_t \qquad (1)$$

where $k = 1, \ldots, t$, and $e_t$ is an uncorrelated random variable for a series with unconditional variance $\sigma^2_t$ and zero mean. Each interval variance is denoted by $\sigma^2_j$, $j = 0, 1\ldots, TN$, where TN is the total number of changes in variances in T observations. By allowing $1 < v_1 < v_2 < \ldots < v_{TN} < T$ equal to the set of breakpoints, the variance is defined by:

$$\sigma^2_t = \sigma^2_0 \; 1 < t < v_1 = \sigma^2_1 \; v_1 < t < v_2 \ldots = \sigma^2_j \; v_{TN} < t < T \qquad (2)$$

where $j = 0, 1, 2, \ldots., NT$.

$D_k$ is defined as:

$$D_k = \frac{C_k}{C_T} - \frac{K}{T} \qquad (3)$$

where $D_0 = D_T = 0$ and $C_T$ is the residual sum square of the whole sample period.

If there are no changes in variance for the whole sample period, then $D_k$ will oscillate around zero. However, if there is more than one shift in the variance, $D_k$ will move from zero. The critical value that defines the upper and lower limits of the drift under the stationary variance of null hypothesis will determine the significant change in the series variance. If, by any means, the maximum of the absolute statistic value $D_k$ is greater than the critical value, the null hypothesis of no sudden change will be rejected (Sayani 2011).

Let $k^*$ be the value of k at which max $k \, | D_k |$ is attained, and if the $\max_k = \sqrt{\left(\frac{T}{2}\right)} * |D_k|$ exceeds the critical values, then $k^*$ is taken as an estimate of the change point. The term $\sqrt{\left(\frac{T}{2}\right)}$ is used for standardised distribution (Fernández 2020). The critical value of 1.358 is the 95th percentile of asymptomatic distribution of $\max_k = \sqrt{\left(\frac{T}{2}\right)} * |D_k|$. Therefore, upper and lower boundaries can be set at $\pm 1.358$ in the $D_k$ plot.

The ICSS algorithm is an iterative approach because the process must be repeated over all sub-samples to identify the structural break and multiple change points. We use the ICSS algorithm to improve the volatility analysis. Using the ICSS algorithm in each precious metal series helps us identify any structural beaks in the time series data and determine the breakpoints during the sample period (Fernández 2020). If breakpoints exist, it makes it difficult to include them in the GARCH equation.

To address this issue, we first implement the standard GARCH model to filter the series for heteroskedasticity. We then conduct variance analysis to check for residuals. After using GARCH, we ran the ICSS on the errors obtained from the GARCH model to obtain a reasonable number of breakpoints for variance analysis. We then used a modified GARCH model, using dummy variables to rectify structural instability. Finally, we compared the results of the standard GARCH model and the modified GARCH model using dummy variables for the rectified breakpoints. This methodology has been widely used in previous studies that explore price volatility (e.g., Aggarwal et al. 1999; Kang et al. 2009; Malik et al. 2005).

*3.2. The GARCH Model with Sudden Changes in Variance*

After identifying the change points in the variance for the time series data of all four precious metals, we estimate the GARCH model with and without sudden changes in variance. A univariate GARCH (1,1) model defines the case for sudden changes in variance as follows:

$$Y_t = \mu + e_t \tag{4}$$

$$g_t = \omega + d_1 D_1 + \ldots + d_n D_n + \alpha e^2_{t-1} + \beta g_{t-1} \tag{5}$$

where $Y_t$ is the price of one of the four precious metals, $e_t \mid L_{t-1} \sim N(0, g_t)$, and $g_t$ is the conditional variance that is subject to *n* exogenous events that cause sudden changes in it and are indicated by the Ds.

In this paper, we adapt the GARCH (1,1) model of Equations (4) and (5) to a bivariate analysis of metal prices. More specifically, the co-movement of changes of metal prices are analysed between gold-silver, gold-platinum, gold-palladium, silver-gold, platinum-gold, palladium-gold, palladium-platinum, palladium-silver, platinum-silver, platinum-palladium, silver-palladium, and silver-platinum.

The below VAR model is used to model the co-movement of the changes of the metal prices.

$$Py_t = c_0 + \sum_{i=1}^{m} ai \, Px_{t-i} + yet \tag{6}$$

$$Px_t = c_0 + \sum_{i=1}^{m} ai \, Py_{t-i} + e_{xt} \tag{7}$$

where $Py_t$ is the change in the price of one of the precious metals, with $Px_t$ being that of another, namely, $Pw_t = \ln(P^{Pw}_t) - \ln(P^{Pw}_{t-1})$. The conditional variances of the error terms in (6) and (7) are modelled according to (5), which modifies the GARCH model by incorporating changes in regime shifts identified by the ICSS algorithm. Volatility persistence (i.e., $\alpha + \beta$) is expected to be smaller than the standard GARCH model.

There will be continuous compounded precious metal returns in this paper, calculated by the first difference in natural log. The literature identified some significant features of precious metals' price volatility, such as asymmetry, fat tails, and long memory (Naeem et al. 2019).

The GARCH (1,1) model of (5) is modified to take into account the effects of precious metal depreciation on each other via the mean returns' equations (Equations (6) and (7)) as well as via volatility. These depreciation effects are modelled as:

$$g_{wt} = C_0 + d_1 D_1 + \ldots + d_n D_n + A \, USDJ_t + \alpha e^2_{wt-1} + \beta g_{wt-1} + \Lambda (Pz_{t-1})^2 \tag{8}$$

where w and z denote any two of the four precious metals prices and the US Dow Jones is denoted by USDJ.

To identify the existence of linear or non-linear dependencies in residuals and the normality of the residuals, we use the Jarque–Bera, Bollerslev–Wooldridge robust t-statistics, and the Ljung–Box (LB) statistics on the standardised residuals for the GARCH and EGARCH models. Finally, we use the ARCH LM residual test to test if there is an additional ARCH in the standardised residuals. We expect that, after specifying the variance equation correctly, there would not be any ARCH left in the standardised residuals.

*3.3. The ICSS and GARCH Model Procedures*

We combined the GARCH model and the ICSS algorithm to improve the analysis of the volatility behaviours of the three different financial markets. Dubois and Bacmann (2002) suggested that the behaviour of the ICSS algorithm is uncertain when there is heteroscedasticity. The authors claimed that one way to work around this issue is to filter the return series using the GARCH (1,1) model, by using the ICSS algorithm to the standardised residuals. In their study, Dubois and Bacmann (2002) found that structural

breaks in unconditional variance are less frequent than previously shown. We followed Dubois and Bacmann's (2002) suggestions, using the steps outlined below.

(1)  First, to identify the possible breakpoint, we used the ICSS algorithm in each series. The results identify a vast number of structural breakpoints in the series but, more interestingly, if the breakpoints happen at different times, this means that it would be difficult to include them in the GARCH model.

(2)  Second, to remove the problem in (1), we used the GARCH model with the objective of filtering the series for the existence of heteroscedasticity and calculated the residuals from the variance analysis.

(3)  In the third step, we ran the ICSS algorithm on the error terms obtained from the GARCH model. This process allows us to obtain a reasonable number of structural breakpoints that could be used in the analysis of variance.

(4)  The fourth step involved the creation of a new GARCH model including the dummy variables to correct for the previous model's structural instability.

(5)  The final step compares the results obtained from the GARCH model, both with dummy variables (breakpoint corrections) and without the correction of structural breaks.

## 4. Empirical Results

### 4.1. Summary Statistics

This section focuses first on obtaining the descriptive statistics of the time series data for the precious metal markets. Some preliminary diagnostic tests are also performed, which include normality test and unit root tests. This will provide us with the characteristics of the dataset. Second, we separately conduct a unit root test for the time series of all four precious metals. The descriptive statistics of precious metal returns reveal a common trend; the four markets (gold, silver, platinum, and palladium) exhibit positive small mean values. For the standard deviations, gold is the most volatile, with a coefficient of 4.79%; silver is less volatile, with a coefficient of 0.04%; platinum is 4.46%; and palladium is 2.56% (see Appendix A Table A1). Table A1 shows the skewness and kurtosis results that indicate that precious metal returns are negatively skewed and leptokurtic. The Jarque–Bera (JB) test rejects the hypothesis of a normal distribution of precious metal returns. The results of the unit root tests indicate that we reject the null hypothesis of a unit root, indicating that the return of the precious metal price series is I(0) (see Appendix A Table A2).

### 4.2. ICSS Break Points

The ICSS algorithm proposed by Inclán and Tiao (1994) is often used together with the GARCH model to analyse the volatility relationship between variables, e.g., see (Kang et al. 2011; Ewing and Malik 2013). Initially, we ran an algorithm on the original time-series data to identify the structural breakpoints to helps us determine when and whether there is a significant change in our data. After running the algorithm, we found that the number of sudden changes in the time-series data was quite high and there was no common breakpoint between the sudden changes in volatility among the precious metal returns. It is a major problem to introduce all the breakpoints in the variance equation. The other problem is to finalise the number of observations needed as breakpoints. One must be careful when eliminating the number of observations from the dataset because this process may generate its own breakpoint; eliminated observations create a gap (or a jump) between observations. The last observation is considered the point where volatility disappears. The purpose of eliminating a number of observations from the dataset is that, with the large number of data points, it is not practical to explore all possible combinations of breakpoints to determine the best fit model.

To overcome this problem, we used the GARCH (1,1) framework with the mean equation for each of the precious metals under analysis. This enables us to evaluate how the number of rapid changes in volatility of precious metals has been reduced. Additionally, it helps tackle the problem of receiving a varied number of days on volatility jumps.

### 4.3. Volatility Results

#### 4.3.1. Gold Analysis

The GARCH (1,1) model results for gold are presented in the first two columns in Table 1. For this analysis, we incorporated the US Dow Jones stock market influence on precious metals to investigate the influence of international financial markets on precious metal markets. When the simple GARCH (1,1) model is used without the correction in structural breaks, the results show a significant negative relationship between the US Dow Jones and gold market (A). In contrast, when a dummy variable was introduced to correct for the structural breaks, the results reveal that the stock market has little or no impact on the gold market. This implies that the GARCH model with dummies tends to correct the level of volatility persistence. These results confirm that the precious metal markets are strong investment options. With regard to volatility persistence, the sum of the coefficients of GARCH is 1, which suggests extreme persistence in volatility in both GARCH (1,1) and GARCH (1,1) with dummies.

**Table 1.** Stability of precious metal prices under the influence of the Dow Jones industrial and structural shifts.

|  | Gold | | Silver | | Platinum | | Palladium | |
|---|---|---|---|---|---|---|---|---|
|  | GARCH (1,1) | GARCH (1,1) with Dummies | GARCH (1,1) | GARCH (1,1) with Dummies | GARCH (1,1) | GARCH (1,1) with Dummies | GARCH (1,1) | GARCH (1,1) with Dummies |
| $c_0$ | 0.0000 | 0.0000 | 0.0000 | 0.0000 | 0.0003 | 0.0002 | 0.0000 | 0.0001 |
|  | (0.7698) | (0.656) | (0.858) | (0.912) | (0.178) | (0.236) | (0.164) | (0.369) |
| A | −0.0354 ** | −0.0043 | 0.0136 | 0.0075 | 0.0510 * | 0.0432 ** | 0.066 * | 0.0032 ** |
|  | (0.015) | (0.763) | (0.533) | (0.729) | (0.004) | (0.018) | (0.012) | (0.0164) |
| Λ | 0.0219 * | 0.0159 ** | 0.0334 * | 0.0319 * | 0.0264 * | 0.0396 * | 0.0333 * | 0.0222 * |
|  | (0.000) * | (0.012) | 0.0000 | (0.002) | 0.0000 | 0.0000 | 0.0000 | 0.0000 |
| $\alpha$ | 0.0504 * | 0.0553 * | 0.0587 * | 0.0563 * | 0.1231 * | 0.1439 * | 0.1381 * | 0.1597 * |
|  | (0.000) * | 0.0000 | 0.0000 | 0.0000 | 0.0000 | 0.0000 | 0.0000 | 0.0000 |
| $\beta$ | 0.9520 * | 0.9843 * | 0.9389 * | 0.9512 * | 0.8709 * | 0.7584 * | 0.8891 * | 0.0636 * |
|  | 0.0000 | 0.0000 | 0.0000 | 0.0000 | 0.0000 | 0.0000 | 0.0000 | 0.0000 |
| $\alpha + \beta$ | 1.0000 | 1.0000 | 0.969 | 0.989 | 0.981 | 0.988 | 0.1 | 0.99 |

* 1% significance level, ** 5% significance level. $\alpha$ is the coefficient for previous shocks and $\beta$ is the coefficient for persistence.

#### 4.3.2. Silver Analysis

The GARCH (1,1) silver results are presented in the third and fourth columns in Table 1. The results show an insignificant relationship between the US stock market returns and silver returns. These results are confirmed by GARCH (1,1) with dummies, which show that silver has an insignificant relationship with the US Dow Jones stock market. This situation reflects that, in general, when the US stock market is appreciating, there is no trend of increasing returns for the silver market.

For the variance estimates, it can be concluded that for both models, the GARCH parameters of volatility persistence have a coefficient close to 1. This suggests a positive correlation coefficient between the US stock market and the silver market.

#### 4.3.3. Platinum Analysis

The GARCH (1,1) model results for platinum are presented in the fifth and sixth columns in Table 1. The results show a significant relationship between the US Dow Jones and platinum returns. These returns are consistent with the GARCH (1,1) model with dummies. For volatility persistence, the overall coefficient of each parameter is significant; the characteristic of the magnitude is reduced for the GARCH with dummies. The sum of the GARCH (1,1) coefficient is close to 1, which indicates extreme volatility persistence for

both GARCH models: the GARCH (1,1) and the GARCH (1,1) with dummies. This confirms the existence of volatility persistence between the US stock market and platinum market.

### 4.3.4. Palladium Analysis

The seventh and eighth columns in Table 1 presents the GARCH model results for palladium. The results show a positive significant relationship between palladium and the US stock market returns, where the coefficient of persistence is 0.88 for the GARCH (1,1) model and 0.06 for the GARCH (1,1) with dummies. For the palladium analysis, the sum of GARCH coefficient is again close to 1, implying the existence of volatility persistence between the US stock market and the palladium market.

### 4.4. Standardised Residuals

The diagnostic tests for the GARCH models on standardised residuals suggest that the residuals are not normally distributed in all cases. For the residual of the ARCH-LM test, the results confirm that the variance equation of both GARCH and GARCH dummy models are specified correctly, since we reject the null hypothesis of remaining ARCH effects in almost all the cases.

To analyse precious metals, this paper used Inclán and Tiao (1994) iterative cumulative sum of squares (ICSS) algorithm to identify the volatility shifts between the precious metal markets (gold, silver, platinum, and palladium) and the US Dow Jones stock market. This paper used the ICSS algorithm to identify sudden changes in variance using each series separately or individually. This allowed us to find a great number of inconsistencies and breakpoints among individual series. Next, we identified common points to find relevant dummy variables that should be included in the GARCH model to analyse for persistence and volatility. To avoid overestimating the breakpoints identified by the ICSS algorithm, we used the GARCH (1,1) model to assimilate standardised residuals. These residuals are used to estimate the relevant volatility jumps in the dummy variables to identify the relationships between precious metal returns and the US Dow Jones stock market.

We next used the GARCH (1,1) model and incorporated the volatility breakpoints identified by the ICSS algorithm. Next, we used the ICSS-GARCH extended model. For the stock market, most coefficients are insignificant, suggesting that shocks in the stock market do not significantly impact the gold market and have only a weak impact on the rest of the precious metals. We also found that the ICSS coefficients are statistically significant after incorporating dummy variables that correct the test results and conditional volatility for sudden shifts.

These results are of great importance to investors because they identify the influence of the US stock market on precious metal markets. Though precious metals are considered a more stable and secure asset class, the independent behaviour of these metals constantly shows an upward trend, even during financial crises. The results also provide evidence of the importance of commodity market exposure in investor portfolios and its independent behaviour, making it an ideal hedging strategy. Lucey and Li (2015) find that at times silver, platinum and palladium act as a safe haven when gold does not. Also when they all act as a safe haven, sometimes precious metals are stronger havens than gold. Analysing the linkages between precious metal prices and the stock market is of particular interest to financial players (Adrangi et al. 2003; Lucey and Tully 2006). Baur and Lucey (2010) conducted a study in the US, the UK, and Germany to identity the relationship between gold prices and stock markets. The authors found gold as a safe haven in these countries after examining the negative shocks affecting the countries' stock markets.

In the light of these results, though global economies are going through financial market uncertainty because of recent events such as the COVID-19 pandemic, political instability, and the introduction of technology advancements, even developed markets appear to be less attractive to investors. In contrast, commodity markets appear as zones of a high level of interest because of its weak correlation with other financial markets. Indeed, precious metal markets need to be investigated carefully and in more detail to capture their

hedging potential. Precious metal markets exhibit price stability properties and can be used for portfolio diversification since the results reveal the undeniable detachment of precious metals from stock markets over the past 20 years. The results will also be of interest to investors who can use precious metals to hedge their portfolios from potential losses.

## 5. Conclusions

The shifts and breaks in volatility were identified using the ICSS algorithm of Inclán and Tiao (1994) to detect multiple breaks in gold, silver, platinum, and palladium.

(a)     With regard to the price stability properties of precious metals, our results suggest that depreciation in precious metals under both stable and volatile market conditions tends to increase market volatility and decrease mean stock returns. Additionally, the existence of volatility persistence among precious metal returns is confirmed by our results. In addition, the results suggest that financial market instability has a weak effect on precious metals. In terms of volatility spillover among precious metals, there is strong evidence of unidirectional volatility spillover, running from gold to the other precious metal markets. There is weak evidence of bidirectional volatility spillover, but the results are mixed across the study period. The asymmetric spillover effect findings endorse the idea that bad news has a greater effect on price volatility than good news.

(b)     Our empirical findings show that though there is no evidence of short-term co-movement among the precious metals, except for palladium, there is certainly long-term linkages in all precious metal markets. These findings imply that palladium might not be a good option for portfolio diversification.

(c)     For gold, most breaks were post the 1997 Asian Financial Crisis and post the 2008 Global Financial Crisis (see Column 1 in Table A4). In general, gold showed evidence of volatility breaks after the 2010 Eurozone Crisis. This implies the absence of volatility impacts on gold prices during the major financial instability environments.

(d)     For silver, most volatility breaks were post the 2010 Eurozone Crisis (see Column 2 in Table A4) so are not directly linked to the crises. This suggests silver exhibits independent price movements during the major financial crises.

(e)     For platinum, all the break points in volatility were around the 1997 Asian Financial Crisis, the 2008 Global Financial Crisis, and pre the 2010 Eurozone Crisis (see Column 3 in Table A4). This implies that platinum prices were highly impacted by financial instability and might not be a stable option for investment portfolio diversification.

(f)     For palladium, a weak impact can be seen during the three financial crises and a major impact by factors and drivers recently, which can be explored in future research to identify the factors outside the financial crises period (see Column 4 in Table A4).

(g)     Additionally, we investigated the evidence of consistency of the correlation among precious metals. We found significant, positive correlation among the precious metals. The strongest correlation was between gold and silver. Strong correlation between precious metals reduces their diversification ability in investment portfolios.

(h)     Our findings provide a better understanding of the precious metal markets and are helpful for investors and portfolio managers to adapt to the market behaviour in terms of volatility and price fluctuations.

Given the significance of the precious metals' market, there are studies that investigate price dynamics of the precious metals with other assets linkages such as Batten et al. (2010), Mighri et al. (2022), and Umar et al. (2021). This paper's major contribution is the analysis of the price stability properties of the precious metal markets among the precious metals. This paper adds to the literature on the relationships and co-movements among the precious metals, since there is a surprising lack of research dealing with co-movements among precious metals during the financial market instability. This paper also highlights the extent of precious metals' returns co-movements along with the most stable metals in terms of price fluctuation.

This paper considers the volatility effects that exist in precious metals by covering numerous data points and investigating the directions of the impacts among the metals. Another contribution is our methodology using the ICSS algorithm on the GARCH (1,1) standardized residuals with precious metals' breakpoints included in the model. The use of the ICSS algorithm on the GARCH (1,1) standardized residuals is another contribution. To the best of our knowledge, there is no evidence of this technique being implemented before to analyse volatility among stock and precious metal market returns.

Finally, we acknowledge the limitation of the ICSS in that it depends on a normality assumption on financial returns, although we applied it to the pair-wise GARCH residuals of the returns which are regarded to be essentially Gaussian. Our sample covers the various financial upheavals and extreme circumstances which could render the normality assumption invalid and, therefore, the ICSS error-prone. This will be our future research where we will use models such as GJR-GARCH or MARS splines to address this limitation of the ICSS.

**Author Contributions:** Conceptualization, S.F. and C.G.; methodology, S.F. and B.H.; software, S.F.; validation, B.H.; formal analysis, S.F.; investigation, S.F. and B.H.; resources, S.F.; data curation, S.F.; writing—original draft preparation, S.F.; writing—review and editing, B.H.; visualization, S.F.; supervision, C.G. and B.H. All authors have read and agreed to the published version of the manuscript.

**Funding:** This research received no external funding.

**Data Availability Statement:** The data were sourced from Bloomberg.

**Conflicts of Interest:** The authors declare no conflict of interest.

## Appendix A

**Table A1.** Descriptive Statistics of the Precious Metals.

| Type of Test | Gold | Silver | Platinum | Palladium | US |
|---|---|---|---|---|---|
| **Mean** | 0.000234 | 0.000365 | 0.000448 | 0.000189 | 0.000358 |
| **Std. Dev.** | 0.008841 | 0.018409 | 0.014598 | 0.010887 | 0.010442 |
| **Skewness** | −0.104 | −0.545431 | −0.648763 | −0.199836 | −0.2742 |
| **Kurtosis** | 10.305 | 1.084 | 1.758 | 6.123 | 7.532 |
| **Jarque-Bera** | 7819 | 8847 | 32455 | 1436 | 2850 |

**Table A2.** The Results of the Unit Root Tests of Precious Metal Prices.

| Type of Test | ADF | | PP | |
|---|---|---|---|---|
| **Precious Metal** | **Levels** | **1st Diff.** | **Levels** | **1st Diff.** |
| XAU | −0.80776 | −7.304936 * | −0.738865 | −32.768 * |
| XAG | −1.586799 | −6.859481 * | −1.584773 | −31.85951 * |
| XPT | −1.923293 | −11.45623 * | −1.790472 | −31.71039 * |
| XPD | −0.693453 | −13.01497 * | −0.633484 | −31.92204 * |

* 1% significance level, XAU: Gold Price, XAG: Silver Price, XPT: Platinum Price, XPD: Palladium Price.

**Table A3.** Diagnostic of the GARCH (1,1) Residuals.

| Precious Metal Pair | Gold-Palladium | Gold-Platinum | Gold-Silver | Palladium-Platinum | Palladium-Silver | Platinum-Silver |
|---|---|---|---|---|---|---|
| LB(20) | 14.68 (0.026) | 15.13 (0.924) | 12.79 (0.953) | 18.75 (0.704) | 15.46 (0.050) | 25.43 (0.016) |
| LB$^2$(20) | 17.91 (0.659) | 35.37 (0.034) | 17.19 (0.725) | 13.78 (0.987) | 3.58 (1.00) | 9.9 (0.98) |
| ARCH-LM | 0.92 (0.7) | 1.66 (0.08) | 0.94 (0.76) | 0.6 (0.98) | 0.18 (1.00) | 0.5 (0.82) |

**Table A3.** *Cont.*

| Precious Metal Pair | Palladium-Gold | Platinum-Gold | Silver-Gold | Platinum-Palladium | Silver-Palladium | Silver-Platinum |
|---|---|---|---|---|---|---|
| LB(20) | 15.14 (0.608) | 24.45 (0.367) | 21.63 (0.52) | 22.18 (0.457) | 34.23 (0.061) | 12.13 (0.972) |
| LB$^2$(20) | 3.44 (1.00) | 13.81 (0.876) | 13.55 (0.948) | 7.64 (0.983) | 11.05 (0.815) | 15.79 (0.83) |
| ARCH-LM | 0.24 (1.00) | 0.69 (0.82) | 0.68 (0.85) | 0.39 (0.89) | 0.69 (0.92) | 0.91 (0.71) |
| Cross Products | | | | | | |
| LB(20) | 17.51 (0.454) | 16.9 (0.623) | 12.35 (0.807) | 26.54 (0.255) | 33.63 (0.032) | 26.76 (0.184) |
| LB$^2$(20) | 0.99 (1.00) | 21.68 (0.466) | 2.98 (1.00) | 7.76 (0.893) | 26.24 (0.169) | 11.79 (0.91) |

**Table A4.** Precious Metal Prices and the US Dow Jones Break Points.

| Gold Series | Silver Series | Platinum Series | Palladium Series | US Dow Jones |
|---|---|---|---|---|
| 9 May 1997 | 29 March 1997 | 9 May1997 | 29 March 1995 | 14 December 1995 |
| 4 March 1997 | 9 May 1997 | 21 September 1998 | 9 May 1995 | 8 January 1996 |
| 11 February 1998 | 2 October1997 | 10 December 1998 | 2 October 1995 | 29 November 1999 |
| 28 May 1998 | 6 December1998 | 18 December 1998 | 2 December 1996 | 14 October 2000 |
| 21 September 1998 | 29 March 1999 | 22 January 1999 | 8 December 1997 | 7 March 2009 |
| 10 December 1999 | 9 May 1999 | 27 January 1999 | 16 December 1997 | 18 October 2012 |
| 18 December 1999 | 2 October2000 | 23 September 1999 | 29 May 1998 | 4 January 2015 |
| 22 January 2000 | 6 January2001 | 18 November 1999 | 15 March 1999 | 13 May 2015 |
| 27 January 2000 | 29 March 2001 | 26 January 2000 | 27 September 1999 | 17 May 2017 |
| 23 September 2001 | 9 May 2002 | 29 February 2000 | 30 September 1999 | 7 June 2017 |
| 18 November 2001 | 2 October2003 | 7 August 2000 | 21 June 2000 | - |
| 26 January 2002 | 6 January2004 | 13 July 2001 | 1 January 2001 | - |
| 29 February 2002 | 28 October 2005 | 16 October 2001 | 14 May 2001 | - |
| 7 August 2002 | 5 May 2006 | 3 June 2001 | 14 August 2002 | - |
| 13 July 2003 | 19 June 2006 | 9 April 2004 | 14 July 2003 | - |
| 16 October 2003 | 30 October 2008 | 17 May 2004 | 2 January 2004 | - |
| 3 June 2003 | 16 November 2008 | 7 December 200 | 12 April 2004 | - |
| 9 April 2004 | 21 November 2008 | 28 October 2005 | 10 May 2004 | - |
| 17 May 2004 | 14 March 2009 | 5 May 2006 | 8 March 2005 | - |
| 7 December 2004 | 22 January 2010 | 19 June 2006 | 5 December 2005 | - |
| 28 October 2005 | 14 March 2010 | 30 October 2006 | 14 April 2006 | - |
| 5 May 2006 | 28 May 2011 | 16 November 2006 | 14 June 2006 | - |
| 19 June 2006 | 21 September 2011 | 21 November 2006 | 3 October 2006 | - |
| 30 October 2008 | 10 December 2011 | 14 March 2007 | 25 February 2008 | - |
| 16 November 2008 | 22 January 2012 | 26 September 2007 | 19 March 2008 | - |
| 21 November 2008 | 23 September 2012 | 15 November 2007 | 14 March 2009 | - |
| 14 March 2009 | 18 November 2012 | 22 January 2008 | 26 September 2009 | - |
| 26 September 2009 | 27 January 2013 | - | 15 November 2009 | - |
| 15 November 2009 | 29 February 2013 | - | 22 January 2010 | - |
| 22 January 2010 | 7 August 2014 | - | 4 March 2010 | - |
| 4 March 2010 | 5 May 2015 | - | 4 March 2011 | - |
| 28 May 2011 | 19 June 2015 | - | 21 October 2011 | - |
| 21 September 2011 | 30 October 2015 | - | 10 December 2011 | - |
| 10 December 2011 | 16 January 2016 | - | 22 January 2012 | - |
| 22 January 2012 | 21 October 2016 | - | 23 October 2012 | - |
| 23 September 2012 | 2 March 2017 | - | 18 November 2012 | - |
| 18 November 2012 | 26 September 2017 | - | 27 February 2013 | - |
| 27 January 2013 | 15 February 2018 | - | 29 February 2013 | - |
| 29 February 2013 | 25 June 2018 | - | 7 August 2014 | - |
| 7 August 2014 | 30 October 2018 | - | 5 May 2015 | - |
| 5 May 2015 | 19 March 2017 | - | 19 June 2015 | - |
| 19 June 2015 | 19 March 2018 | - | 30 October 2015 | - |
| 30 October 2015 | - | - | 16 January 2016 | - |
| 16 January 2016 | - | - | 21 September 2016 | - |
| 21 September 2016 | - | - | 14 April 2017 | - |
| 14 March 2017 | - | - | 26 September 2017 | - |
| 26 September 2017 | - | - | 19 February 2018 | - |
| 15 January 2018 | - | - | 25 June 2018 | - |
| 25 June 2018 | - | - | 4 August 2018 | - |
| 30 October 2018 | - | - | - | - |

Break points calculated using the ICSS algorithm for each of the series. The ICSS algorithm for the individual series overestimates the number of points where sudden changes in volatility occur. The same situation was found in the case of the precious metal indices. Each series presents a different number of break points that happened on different days during the period of analysis. In order to obtain a common number of breakpoints we used the standardised residuals of the GARCH (1,1) model. This allowed us to determine the appropriate number of break points to improve our variance equation.

**Table A5.** GARCH (1,1) Residuals Break Points for Four Precious Metal Pair with US Dow Jones.

| Gold-US | Silver-US | Platinum-US | Palladium-US |
|---|---|---|---|
| 28 March 1999 (obs.63) | 29 September 1999 (obs.1139) | 31 May 1996 (obs.371) | 29 September 1999 (obs.1239) |
| 29 March 2002 (obs.64) | 7 May 2001 (obs. 657) | 10 February 1997 (obs.552) | 30 December 2002 (obs.522) |
| 30 December 2002 (obs.522) | 7 July 2003 (obs.942) | 27 August 2002 (obs.198) | 5 September 2010 (obs.1002) |
| 28 May 2010 (obs.72) | - | - | 3 August 2015 (obs.153) |
| 5 September 2010 (obs.1210) | - | - | 5 December 2016 (obs.153) |
| 12 September 2010 (obs.1210) | - | - | - |

The ICSS algorithm using the GARCH (1,1) standardised residuals have the advantage of reducing the number of points where sudden changes in volatility occurs. They also have the quality of providing a common break point for our mean equation, allowing us to reduce the number of dummy variables that should be introduced into the GARCH (1,1) variance equation.

## Note

[1]   A safe-haven asset is an asset that holds its value in adverse market conditions. Such an asset offers investors the opportunity to protect wealth in the event of negative market conditions Baur and McDermott (2010).

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
