# Peer review of "Price Stability Properties and Volatility Analysis of Precious Metals: An ICSS Algorithm Approach"

_jrfm, doi:10.3390/jrfm15100465_

Round 1

Reviewer 1 Report

Dear Authors,

The author needs to clarify the new contribution of the research in the introduction. It is necessary to clearly state the new and motivating points of the article.

The literature review should be placed after the missing section. Authors need to update recent studies. And point out the missing point to carry out this study. The author should have a literature review to compare the results of previous studies conducted in the same research context.

The author needs to adjust the summary to be concise and not to exceed the specified word count.

The author needs to adjust the form of tables, and decimals, ...

The author needs to separate the literature review and introduction sections.

The author should discuss the method clearly and emphasize its superiority.

The author needs to analyze the results and compare them with the stated background theory.

I hope my comments may help you in developing the paper.

Author Response

Dear Reviewer,

Thank you for your valuable suggestions and comments.

We have re-written parts of the introduction to highlight the new contribution of our research, namely, our research presents the evidence that while financial stock markets have experienced instability some precious metal markets have remained relatively calm, especially the gold market.  Also new is that we used the ICSS to address possible shifts in volatilities and to evaluate their effects on price instabilities.

We have separated the literature review and introduction sections and adjusted part of the writings to explain our method.  The discussion of the empirical results are also modified to better align with the background of the research.

Reviewer 2 Report

File is attached.

Author Response

Dear Reviewer,

Thank you for your valuable suggestions and comments.

We have incorporated your comments in our paper to indicate the limitation of our research.  We have also indicated that our future research will address this limitation by using the models you suggested.

Yours sincerely,

Authors

Reviewer 3 Report

This paper investigates the price stability properties of precious metals during the 1997 Asian Financial Crisis, 2007-2008 Global Financial Crisis and 2010 Eurozone Crisis. My major comments are listed below.

1)    The sample does not include a recent period, such as the COVID-19, Brexit and/or US presidential election 2019. Please justify why more novel samples are not considered. The investigated dataset is at least 10 years ago, which may not be sufficient to motivate the usefulness of the results.
2)    The ICSS method is also not a novel methodology. At the very least, the authors should consider at least one alternative structural change model to check the robustness of their results. Such models may include regime switching GARCH, and/or time-varying coefficients GARCH.
3)    Similarly, it is more interesting to consider a multivariate model, rather than individually model the four autocorrelated precious metal volatilities. This enables the readers to explore the spillover effects and related features.
4)    Minor issues: tables 1-4 may be combined as one, and you may only need one subsection (4.3), rather than four subsubsections.

Author Response

Dear Reviewer,

Thank you for the suggestion to augment the sample to accommodate events like COVID-19, etc..  We plan to do a second paper which will include an up-to-date sample.  For the present study, we just wished to focus on the financial upheavals characterised by the AFC, GFC and EC.

The ICSS has its limitation in that it requires the time series process to be Gaussian.  Financial returns are generally regarded Gaussian, however, this requirement may not be met given that there were events like GFC during the period of our sample.  We have acknowledged this limitation in the revised version of our paper. 

We agree that a multivariate GARCH would be more interesting.  The reason why we did use such a model was because our main interest was in the interconnectedness between each of the precious metals’ markets and the stock market.  A secondary reason was that computation was a lot easier for a bivariate GARCH than a 5-dimensional GARCH.

Thank you we have merged the four tables into one.

Yours sincerely

Authors

Round 2

Reviewer 2 Report

I am fine.

Reviewer 3 Report

I thank the authors for revising their paper according to my comments. I recommend publication as is.